# Artificial Reproductive Technology Use and Family-Building Experiences of Female Adult Childhood Cancer Survivors: A Qualitative Study

**DOI:** 10.3390/curroncol32070369

**Published:** 2025-06-25

**Authors:** Selena Banser, A. Fuchsia Howard, Sally Thorne, Karen J. Goddard

**Affiliations:** 1Vancouver Coastal Health, Plastic and Reconstructive Surgery, 520 West 6th Ave, Vancouver, BC V5Z 4H5, Canada; sbanser7@mail.ubc.ca; 2School of Nursing, The University of British Columbia, T201-2211 Wesbrook Mall, Vancouver, BC V6T 2B5, Canada; sally.thorne@ubc.ca; 3Women’s Health Research Institute, BC Children’s & Women’s Hospital, 4500 Oak Street, Vancouver, BC V6H 3V4, Canada; 4BC Cancer, 675 West 10th Ave, Vancouver, BC V5Z 0B4, Canada; kgoddard@bccancer.bc.ca; 5Faculty of Medicine, The University of British Columbia, 317-2194 Health Sciences Mall, Vancouver, BC V6T 1Z3, Canada

**Keywords:** fertility, oncology, assisted reproduction, in vitro fertilization, pregnancy, cancer survivor

## Abstract

Female adult childhood cancer survivors often face challenges with fertility due to their cancer treatments. While some may turn to assisted reproductive technology or other family-building options, there is limited research on their experiences. This qualitative study explored the challenges faced by female survivors in a western Canadian province as they navigated family-building options. Interviews with 15 participants revealed five key challenges: unexpected impaired fertility, grief and identity struggles, unsupportive healthcare experiences, the complexities of adoption and international family-building, and significant financial burdens. These findings highlight the complex and emotional nature of fertility and family-building for survivors and point to important gaps in healthcare services. The study emphasizes the need for integrated informational, psychosocial, and financial support related to family-building after cancer.

## 1. Introduction

Childhood cancer survivors have commonly expressed a desire to have children in the future [1], with having children an important determinant of quality of life among survivors [2,3]. Yet, for female adult childhood cancer survivors (ACCSs), this desire can be hampered by compromised reproductive function. Sub-fertility or infertility can result from cancer treatments that damage the reproductive system, including the ovaries, uterus, and hormonal pathways that control puberty, menstruation, and ovulation [4,5,6]. Particularly harmful treatments include high doses of alkylating or heavy metal chemotherapeutics [7], radiation therapy (RT) to the pelvis, spine, brain, or total body, surgery affecting reproductive organs, and stem cell transplant [5,6,8]. Both chemotherapy and RT can lead to premature ovarian insufficiency or diminished ovarian reserve [6].

Given the risks of sub-fertility and infertility, there is a general consensus and guidelines that oncologists should have a conversation about fertility and fertility preservation options with all cancer patients of reproductive age, or with the parents of children, and refer at-risk patients to fertility specialists early during primary cancer treatment [9,10,11,12]. However, at the time of cancer treatment, there are few fertility preservation options available to female pediatric patients. While ovarian tissue cryopreservation with future re-implantation was previously considered experimental, current guidelines recommend its consideration in carefully selected cases [11,12,13]. For adolescent females who have begun puberty, oocyte cryopreservation is the standard of care for fertility preservation [11,12,13]. However, this may not be feasible before initiating cancer treatments due to the time and coordination of multiple medical teams required in the context of the urgent need to start cancer treatment [10]. Following cancer treatment, routine endocrinology monitoring and testing for females who received gonadotoxic cancer treatment is recommended starting at the age of 13 [14].

When initiating family planning or building, ACCSs who experience subfertility or infertility might seek out assisted reproductive technology (ART) [1,15,16]. ART includes any fertility-related treatments in which eggs or embryos are manipulated, with in vitro fertilization (IVF) the most common procedure [17]. ACCSs who were able to cryopreserve oocytes before treatment may have the option of IVF. Younger survivors who are not ready to have a child, but realize they have a protracted fertility window, can pursue oocyte or embryo cryopreservation for future use, provided they still have high-quality ova. For female ACCSs who can carry a child but suffer from ovarian failure, donor oocytes or donor embryos in conjunction with IVF may be an option. While ARTs offer hope, there is no guarantee of success, and they can be expensive and inaccessible to many [18].

Family-building for infertile female ACCSs might include third-party reproduction where a donor contributes ova, an embryo, or a uterus (in the case of gestational carriers/surrogates) [4]. Traditional surrogacy involves the fertilization of a surrogate’s ova with the intended father’s sperm via intrauterine insemination, whereas a gestational carrier does not use her ova, but rather achieves pregnancy through IVF [19]. The gametes used may be those of both intended parents, a donor oocyte with the intended father’s sperm, the survivor’s oocyte with donor sperm, or a donated embryo [19]. However, in some countries, such as Canada, it is illegal to pay a surrogate. Adoption may be another family-building option, but can be costly and problematic to navigate [20,21].

The direct consequences of uncertain fertility, sub-fertility, and infertility can be devastating for cancer survivors, contributing to anxiety, depression, distress, feelings of loss and grief, disrupted identity formation, and intimate relationship challenges [22,23,24,25,26,27,28,29,30,31,32]. Despite this robust research, there is a dearth of patient-perspective evidence describing the challenges female survivors experience when pursuing ART and family-building options. This evidence is foundational to developing clinical programs and services that meet ACCSs’ unique needs. Further, evidence that is grounded in local healthcare service contexts and the experiences of those interacting with such services is necessary for context-relevant solutions. Thus, the research purpose was to describe the challenges female ACCSs experienced while navigating ART and family-building options in Canada, intending to use what was learned to inform clinical practice.

## 2. Materials and Methods

The protocol for this qualitative, interpretive description [33] study was granted ethics approval by the University of British Columbia Ethics Board (ID#H2003706). This study was performed in line with the principles of the Declaration of Helsinki. We obtained informed consent from all individual participants included in this study.

### 2.1. Setting and Study Participants

This study was conducted in British Columbia (BC), Canada, where the Canadian public healthcare system generally does not fund ART or family-building services. In BC, many childhood cancer survivors are followed by a provincial oncology/hematology/bone marrow transplant program for their survivorship care until they enter young adulthood, after which they transition to their primary care provider for ongoing survivorship care. ACCSs deemed to be at high risk of late effects based on their cancer treatments are transitioned to the provincial Late Effects, Assessment and Follow-Up (LEAF) Clinic. However, the LEAF clinic also provides care to a large number of ACCSs who were initially lost to follow-up after cancer treatment and may not have been receiving risk-based survivorship care.

Purposive sampling through the LEAF clinic was used to recruit ACCSs who were female, English-speaking BC residents, treated for a childhood malignancy at the age of 19 years or younger, currently in remission with no evidence of disease, and had pursued ART or family building, with at least a portion of their experience being in BC. A clinician research team member identified eligible participants and obtained consent for a research team member to make contact and provide additional study information. Individuals were then contacted by a study team member who had no previous relationship with the individual, and those who provided informed consent to participate were enrolled in the study. Only one individual who was contacted by the study team declined to participate in the research, indicating that the subject was too emotional to discuss. Recruitment was from April 2021 to December 2024, and took longer than expected because of the disruption to clinical services and research associated with the COVID-19 pandemic. Our sampling was guided by the concept of information power [34] rather than data saturation. Following interviews with 15 individuals, we determined our data to be high in information power because all participants held characteristics that were highly specific to the narrow aim of the study, and they represented diversity in terms of pediatric cancer diagnoses and treatment modalities, and experiences of ART and family-building.

### 2.2. Data Collection

We conducted one-on-one, semi-structured virtual interviews (via Zoom Video Communications, Inc., San Jose, CA, USA), though we only used audio. The majority of interviews were conducted by S.B., a nursing graduate student with oncology expertise, following training from team members with expertise in qualitative research. An interview guide was created and reviewed by all research team members, with main topics including the challenges, difficulties, surprises, facilitators, and barriers encountered while navigating ART and family-building options. Taking an iterative approach to data collection and analysis, we reflected on emerging insights and patterns from the interviews, interrogated these emerging ideas, and adjusted the interview guide accordingly. While we had question prompts, we encouraged participants to speak freely about their ART-related and family-building experiences and communicate the ideas and thoughts that were most important to them. We wrote field notes following each interview to contextualize the interview, note specific insights from the interview, raise questions that could inform subsequent interviews, and serve as an audit trail. The mean interview time was 41 min, and interviews were audio recorded, transcribed verbatim, de-identified, and reviewed for accuracy. We gathered participants’ self-reported sociodemographic information at the beginning of each interview.

### 2.3. Data Analysis

Following an interpretive thematic approach [33], data analysis began by reading the transcripts multiple times to familiarize ourselves with the data and to identify initial inductive codes. At the stage of initial inductive coding, we identified units of meaning relevant to the research question, using participants’ language where possible, to develop a coding framework. The qualitative data management program NVivo^TM^ version 12 was used to apply the coding framework to the full data set. The codes were reviewed and refined through constant comparison of data within and across codes and across participants. As coding progressed, we grouped and regrouped codes into broader categories and themes reflecting the participants’ experiences in relation to the research question. This interpretive process involved moving between the data and developing themes that remained grounded in the participants’ accounts while also offering interpretive insights, aiming to generate practice-relevant evidence [33]. One author (S.B.) conducted the majority of the initial inductive analysis, creation of the initial coding frame, coding of data, constant comparative analysis, and generation of themes, creating analytic memos throughout the process. However, this process involved team members’ ongoing engagement with the data, analytic discussions, and multiple revisions of the findings.

### 2.4. Rigor

Collectively, our research team had experience and expertise in pediatric and adult cancer treatment, cancer survivorship research and clinical care, cancer-specific late effects, and qualitative research, including Interpretive Description. Our expertise bolstered the study’s credibility and our ability to ask meaningful and clinically relevant questions and engage in thoughtful conversations during data collection, to contextualize the data, and to reflect on and refine preliminary and evolving findings and interpretations. All team members critiqued evolving findings to enhance representative credibility, analytic logic, and interpretive authority. We also engaged in ongoing reflexivity, considering how our professional backgrounds, clinical experiences, and positionalities shaped the questions we asked, how we interpreted participants’ narratives, and the meaning we generated from the data. One team member (S.B.) maintained an audit trail, including reflexive and analytic notes, draft analysis materials, and documents of methodological decisions.

## 3. Results

A total of 15 female ACCSs who ranged in age from 25 to 48 years (mean age 35.6 years) participated. All identified as heterosexual, with 13/15 indicating they were Caucasian. The average age at cancer diagnosis was 9.5 years, with a range of cancer types. All were treated with chemotherapy; 10 received RT; 9 had surgery for partial resection; and 2 underwent a bone marrow transplant. Eight pursued IVF with their own ovum, with three becoming pregnant with a subsequent live birth; five pursued IVF with donor ovum, with one successfully carrying a pregnancy to term; two pursued surrogacy and two pursued intrauterine insemination. Several participants used more than one ART. Among the 15 participants, there were a total of 6 miscarriages and 1 ectopic pregnancy, and 5 had children at the time of interview (See Table 1 for sociodemographic characteristics and Table 2 for medical characteristics, and ART and family-building strategies used).

The commentaries of the 15 female ACCSs brought to light five prominent challenges experienced while navigating ART and family-building options, including (1) confronting unexpected, impaired fertility, (2) grieving loss and redefining identity, (3) encountering unsupportive healthcare, (4) exploring alternative paths of adoption and international family-building, and (5) facing financial strain.

### 3.1. Confronting Unexpected, Impaired Fertility

Upon learning of their impaired fertility, all but one participant expressed feelings of surprise and shock, which they attributed to either a lack of information or misconceptions about their fertility status. Survivors commonly indicated they had not been sufficiently informed of their infertility risk, nor had they been offered fertility assessments as part of their medical care until much too late. Some described limited information or discussions about fertility during their cancer treatment, where “*nobody ever said anything to me*”, though some acknowledged they were too young at the time of diagnosis to remember such conversations. Others described instances where their fertility concerns were dismissed at the time of cancer treatments. For example, one participant, diagnosed at age 17, recalled that she and her parents had raised questions about fertility before beginning treatment but that the oncologist reassured them, expressing confidence that fertility would not be affected. Much to the survivor’s dismay, her fertility was severely impacted, and after one cycle of oocyte cryopreservation, only three ova were retrieved.

Following treatment, several survivors described being vaguely aware of the potential for sub-fertility or infertility but had not anticipated these risks to occur at a young age. Many had assumed fertility because they had continued to menstruate as young adults, albeit some with menstrual cycle irregularities. They did not link their irregular menses to infertility risks and were unaware of their early menopause likelihood and shortened fertility window. Most only became aware of their infertility risk when they sought medical advice in preparation for or when trying to conceive. Yet, even then, most did not receive comprehensive information or fertility assessments. For example, one woman was vaguely aware that she might encounter difficulties because she had had a bone marrow transplant. She sought information from her oncologist and endocrinologist at the time in preparation for trying to conceive,


*“We just went in to see [if I might have troubles conceiving], but all they said was ‘oh, you should just try when you’re ready, but try as early as possible.’” (Diagnosed with leukemia at 3, currently 32; 4+ years trying to conceive.)*


Subsequently, her family physician expressed concern based on the advice to try as early as possible, yet no additional assessments were offered. This participant went on to have several unsuccessful attempts using IVF with her oocytes. Another participant who was 19 when diagnosed with cancer described understanding the risks of infertility; however, she also indicated she received little follow-up and no evaluation of her fertility status:


*“I understood the risks. I stopped having my periods during chemotherapy. […] There was a chance that my periods just wouldn’t come back. So, the fact that they did and were relatively regular was a good sign, but it was more of a we’ll see how long it lasts kind of conversation.” (Diagnosed with lymphoma at 19, currently 39; 1 year trying to conceive.)*


She went on to explain that once she was ready to start building a family, she was infertile, and IVF was no longer an option.

As a result of limited awareness of their infertility risk, several survivors suggested they missed the opportunity for early intervention and mental preparation for ART, or even the viability of ARTs. They indicated that they were already in premature ovarian failure when their fertility became a concern, meaning that IVF with their own oocytes was no longer an option. That is, they described missing the opportunity for early fertility assessments, oocyte preservation, and IVF, as depicted by one woman:


*“It was a risk that I would hit menopause early. It wasn’t really described what all that meant, if I’m being honest. I thought IVF was going to be an option. That’s the piece I didn’t quite understand. I always kind of assumed that if I had difficulty conceiving or having kids, IVF would be an option, basically that I could throw money at the problem and it would help, but that wasn’t the case. In hindsight, I would have requested some of the hormone tests earlier. That would have probably painted a better picture of at least showing some decline, and I could have known a little sooner or maybe prepared sooner or changed plans or things like that.” (Diagnosed with lymphoma at 19, currently 39; 1 year trying to conceive.)*


Those who experienced treatment-related premature ovarian failure at a young age indicated they would have appreciated the opportunity to cryopreserve their oocytes before or following treatment, so IVF with their own oocytes would have been an option. However, it appeared that this approach to fertility preservation was not offered in all cases. It was not until after navigating infertility that the survivors understood what fertility preservation meant, and they became acutely aware of the possibilities of fertility preservation if pursued earlier in their adolescent or young adult years. For instance, one survivor who was post-pubertal at the time of cancer treatment questioned whether oocyte preservation could have been an option:


*“I had gone through puberty, so technically, my eggs probably could have been harvested at the time. I don’t know if that was even something that they were doing back then, but I have spoken to other people who had different types of cancer around the same time that I did, like at the same age, and they were able to freeze eggs. So, that just seemed to have been kind of like overlooked for me.” (Diagnosed with a brain/CNS tumor at 14, currently 33; 4+ years trying to conceive.)*


Survivor narratives also highlighted a lack of information or discussions about how their past cancer treatments might impact their ability to carry a pregnancy, resulting in limited awareness of potential complications. Two individuals who pursued IVF were able to achieve pregnancy after several attempts. However, both shared that concerns about their inability to carry to term were only raised by healthcare providers after they became pregnant, not beforehand or during their fertility planning. One explained that she had significant uterine lining irregularities from radiation therapy, while the other had fragments of her pelvis removed during cancer treatment. Neither had been informed of the implications their treatment would have for pregnancy outcomes and both experienced pregnancy loss.

In contrast to the majority of survivors who experienced surprise and shock upon learning of their infertility risk, one survivor whose cancer treatment included a hysterectomy appeared to have understood that pregnancy would be impossible, commenting that:


*“In terms of fertility, I was lucky that it was always talked about in my family, I was well prepared that carrying a child would never be an option for me. And so, with medical advancements, hopefully, by the time I was ready to have a child, surrogacy would be an option. So that’s why at 19 years old, I tried independently to preserve my eggs because I wasn’t sure if I would go through early menopause.” (Diagnosed with rhabdomyosarcoma at 1, currently 34; 4+ years trying to conceive.)*


### 3.2. Grieving Loss and Redefining Identity

The survivors commonly emphasized the enormity of loss and grief they faced upon learning they were at risk of becoming, or had become, infertile. For most, the unexpected nature of their impairment or loss of fertility, especially at a young age, seemed to exacerbate their sense of loss and grief. For example, one survivor characterized her experience of premature ovarian failure at age 16 as “*the most traumatic thing that has happened since I’ve had cancer*”.

The discovery of infertility led all survivors to explore their family-building options, which included hormone therapy, IVF with one’s own cryopreserved oocytes, IVF with donor oocytes, surrogacy with both donor embryos and one’s own embryos, and intrauterine insemination with donor sperm. The survivors spoke of the immense emotional challenges involved with exploring and pursuing such options and their experiences of loss of relationships, miscarriage, ectopic pregnancies, and failed IVF attempts. Several survivors described how loss after loss related to unsuccessful attempts to conceive or build a family contributed to the dissolution of their marriages or intimate relationships. Survivors commonly characterized their experiences of miscarriage as traumatic and accompanied by enduring grief, as exemplified by one woman’s description of her miscarriage after a third IVF attempt using a donor embryo:


*“So, they sent me back down to the emergency room and they did an exam and they told me just like right away, like with the ultrasound and everything that, that I had actually lost the baby six weeks previous. I think I spent a good week in bed after that happened. It was a lot of excitement thinking like, ‘oh, this is actually working, I’m actually going to be able to do it and then to just feel like someone ripped the tablecloth out from underneath me,” (Diagnosed with a brain/CNS tumor at 14, currently 33; 4+ years trying to conceive.)*


Survivor commentaries conveyed profound grief when the loss of fertility was accompanied by awareness of the inability to use IVF because oocyte cryopreservation had not been carried out. They described this as simultaneously losing the ability to have a biological child, to experience pregnancy and carry a child, and to be a mother, potentially. The survivors shared how this deeply emotional, disappointing, and life-changing realization forced them to contemplate their future in a way they had not yet considered.

The survivors described at length how such experiences resulted in feelings of hopelessness, leaving them uncertain as to whether they wanted to continue to pursue motherhood, as with each failed attempt to build their family came a deeper sadness. Participants noted enduring periods of sadness, depression, anger, fear, guilt, isolation, loneliness, and frustration coupled with ongoing grief, which culminated in debilitating mental health for some: “*you don’t see the light at the end of the tunnel*”. Yet, hope and acceptance were also described, though it appeared to be different for each survivor. One woman focused on the role of hope in helping her navigate fertility treatments with various specialists, while another woman found acceptance in realizing she had exhausted all options to conceive and decided to pursue surrogacy:


*“Surrogacy. Deciding, putting at peace that I was never going to get pregnant. I did every diet, I did every supplement, name it, I tried it, I did it. There’s something peaceful in knowing that I gave 110% and it’s time to move on. Once I accepted that, there was a weight lifted off my shoulders and I could be at peace knowing I did everything. It’s time for me to take the weight off my shoulders and move on to something that isn’t reliant on me,” (Diagnosed with Wilms tumor at 2, currently 31; 4+ years trying to conceive.)*


Other survivors indicated that acceptance came with a sense of certainty, prompting some to choose fertility packages that included repeated attempts at surrogacy until a live birth was achieved, at no extra cost per attempt. Others described the acceptance that came with committing to a happy life without children. All women suggested they had come to terms with their losses, albeit after several years of loss and grief.

Survivor’s narratives of impaired fertility and family-building attempts also appeared to reflect a profound sense of disrupted self-identity. They described “*feeling like less of a woman*”. and being a “*disappointment in myself; I felt like my body was failing me*”. For most, it seemed that womanhood, motherhood, and the ability to become pregnant were deeply tied to their identity. Learning of their infertility, or facing failed IVF attempts, miscarriages, or ectopic pregnancies, seemed to trigger feelings of inadequacy and self-doubt. Survivors commonly described no longer feeling like themselves during those difficult moments. One survivor reflected,


*“It took a little bit of my femininity away. […] feeling robbed, or like less of a woman because your inside parts don’t work the way they are supposed to.” (Diagnosed with a brain/CNS tumour at 14, currently 33; 4+ years trying to conceive.)*


Feelings of guilt were also commonly described as impacting self-worth. Some reported excessive guilt at being unable to provide a biological child to their partner. One survivor, following a miscarriage, shared the belief that she had failed to provide a safe home for her child during pregnancy. Others expressed guilt at disappointing their parents, who had hoped for grandchildren.

### 3.3. Encountering Unsupportive Healthcare

Across all survivors’ narratives, healthcare providers were depicted as operating in a system often insensitive to their unique needs, particularly concerning fertility and pregnancy loss. Interactions with non-oncology healthcare providers highlighted a lack of awareness not only of their cancer history but also of the resulting fertility challenges. This lack of awareness was seen to be exacerbated by a fragmented healthcare system where fertility-specific care was not integrated, leading to communication mishaps and missed opportunities for interactions with trusted professionals. Several survivors described instances where healthcare professionals unfamiliar with their cancer history failed to comprehend the complexities of impaired fertility. Further, existing healthcare services were described by all survivors as inadequate or inappropriate for addressing fertility-related challenges appropriately. For instance, one survivor described devastating interactions during a miscarriage, where healthcare providers on a labor and delivery unit seemed ill-prepared to support someone through pregnancy loss.


*“My water blew up. It’s on me, it’s on her, everyone’s soaked with amniotic fluid, and she looks at us, and like the whole world started spinning. You feel like you’re going to die. And she looks at us and the first thing she says is ‘Oh my gosh. Okay guys don’t worry. You’re so young, you’re going to have another baby.’ And I’ll never forget that moment. And I just looked at her, and I was like, ‘No you don’t understand. It is impossible for us to get pregnant; I need this baby.’ I literally felt like I was going to die. After all that, my whole world ended.” (Diagnosed with Wilms tumor at 2, currently 31; 4+ years trying to conceive.)*


She further described the inappropriateness of even being cared for on a labor and delivery unit.


*“Everyone’s having babies and there’s families and balloons and life’s great. And you’re in a room waiting to deliver your dead baby while there are all these celebrations going on around you. It’s just sick.” (Diagnosed with Wilms tumor at 2, currently 31; 4+ years trying to conceive.)*


Because there appeared to be no real link between healthcare services and family-building options, all of the survivors described instances wherein healthcare providers had attempted to be supportive and encouraging, though some indicated that this was done without a full understanding of realistic options. As such, some survivors explained that healthcare providers were quick to recommend adoption as an easy alternative should infertility be a reality. However, the survivors commonly indicated that adoption was highly inaccessible, especially for cancer survivors, and should not be conveyed as an easy alternative. Some of the survivors also felt judged by healthcare providers when they sought alternative approaches to family building that were unavailable in Canada. For example, one woman recalled that a physician refused to review her bloodwork results because they had not received the full treatment file from overseas.

While several of the survivors described their positive interactions with healthcare providers at fertility clinics, they also indicated that providers were not always equipped with knowledge about the survivor’s cancer trajectory or the implications for their fertility. As such, some felt that the rapid speed of fertility treatment consultations and the required regimented treatments were not necessarily personalized with their cancer history in mind. For example, one woman discussed the lack of awareness among the fertility specialists of her complex medical history when she became pregnant with triplets following IVF.

*“The fertility clinic said that we should transfer two embryos, to have a better chance of one of them working. And so, we foolishly listened to that advice, and that was the transfer that worked. But then I became pregnant with triplets.* […] *So, the fertility clinic said ‘congratulations, this is so good’. And I was like, ‘no, this is actually horrible news’. So, I called [oncologist] and said, ‘what do you actually think of this? Because they’re saying that I should just move ahead and go ahead and try to have triplets.’ And she said ‘that would be absolutely terrible for you with your medical history and your advanced maternal age and all the rest of it. You’d be risking your life and probably end up with no baby.’ … So, it was very difficult to have to go through that.” (Diagnosed with Wilms tumor at 7, currently 40; 4+ years trying to conceive.)*

Some survivors also described the fertility clinics as having a “*for-profit*” attitude, contributing to their “*feeling like a number*” and being treated “*as if I was shopping for options*”. One survivor mentioned,


*“I think that they give you hope that they’re going to help you figure out a way to somehow get pregnant. But I just don’t know. Not to sound jaded because I’m very lucky that I had good doctors at the fertility clinics, but it also is business for them.” (Diagnosed with rhabdomyosarcoma at 6, currently 41; 1 year trying to conceive.)*


### 3.4. Exploring Alternative Paths: Adoption and International Family-Building

The survivors highlighted the marked challenges associated with adoption. They described healthcare providers suggesting adoption as a feasible alternative to fertility treatments or if ART was not viable, but this proved misleading. Those who investigated adoption found it generally inaccessible, especially following cancer treatment. One woman described it as “*the hardest option you can choose*”, indicating that cancer survivors are often ineligible to adopt in many countries. Another noted that adoption agencies reject cancer survivors due to concerns about their lifespan, cancer recurrence risk, and ability to provide for a child. Additionally, she shared that waitlists for adoption can exceed 10 years, and the process involves significant financial costs and uncertainties in most countries, including the possibility of not being a successful applicant or having the child removed after placement.

The use of donor oocytes and embryos as well as paid surrogacy are not available services in Canada. Because oocyte removal for fertilization and subsequent IVF was not a viable option for many survivors, six women indicated that they sought ART services outside of Canada, including Europe, the Middle East, the United States, and Ukraine. One woman described the barriers to surrogacy in Canada that led her to pursue international surrogacy:


*“*
*We’ve been looking in Canada to try and find a surrogate for five years, and we came close last year to finding somebody, and then they changed their mind after six months. We just had no leads, and it’s very competitive in Canada because I think you have a lot of international couples that want to come and do surrogacy. And, because you cannot pay the surrogate mother, not that we could afford to, as another issue, but because there’s not that option, it’s really hard to convince somebody to do that.” (*
*Diagnosed with rhabdomyosarcoma at 1, currently 34; 4+ years trying to conceive.*
*)*


While several survivors were able to pursue international family-building options, others indicated that the high costs made this out of reach for them.

### 3.5. Facing Financial Strain

All the survivors spoke about their disappointment at the high expense and limited funding available to assist them with ART or family-building options. They explained that Canadian health insurance covered the cost of some of the hormone medications, but the remainder of ART was not covered. As such, the women indicated that their ARTs cost roughly CAD 20,000–30,000 each, but with costs increasing with each failed attempt and/or additional pregnancy attempt. While medical insurance through their employer offset some of the expense for some, this also presented challenges.


*“I was extremely fortunate that my employer, in [country] had very, very good healthcare and very good health insurance. I worked at an international organization for six years and hated every moment of it, but the reason that I stayed there was because the health insurance was so good, and it was international health insurance. So, it covered 93% of my fertility treatment in BC.*
*” (Diagnosed with rhabdomyosarcoma at 1, currently 34; 4+ years trying to conceive.)*


In addition to the already high costs of pursuing ARTs, survivors living far from fertility clinics incurred significant additional expenses for travel to attend appointments and tests, further adding to the financial burden.

The survivors who sought services outside of Canada indicated that they spent even more than when accessing domestic services; a minimum of CAD 50,000, but in most cases, upwards of CAD 100,000. These high costs were driven by exchange rates with foreign currencies that were much stronger than the Canadian dollar, additional expenses for flights or transportation and accommodation, as well as the costs of importing donor oocytes and embryos.

Whether seeking domestic or international fertility or family-building options, all survivors experienced a significant shortfall in funding to cover costs and described shock when learning of the limited financial assistance available. As a result, some survivors delayed or did not pursue costly ARTs or family-building options, as described by the youngest participant.


*“The woman told me that there was less than 1% chance that I would be able to use the one egg they’d be able to retrieve. And it was $10,000 to do the whole procedure on top of the storage fees. And I asked about compassion medication and cost coverage because I know they do that for women who are going into chemotherapy, and she basically was like ‘Well, that funding is for people going into chemotherapy and for those who can do the retrieval before going through the whole process. There’s nothing available for you.’ I remember I was quite shocked by that, and she just didn’t really seem to care. I ended up not doing anything because $10,000 is a lot of money. It was money that I didn’t have, and it was money that my parents didn’t have.” (Diagnosed with leukemia at 3, currently 25; has not begun trying to conceive.)*


Others described draining their savings accounts, depending on familial financial support, or re-mortgaging their residence.

Thus, the findings described above provide a portrait of the practical and emotional complexities associated with family building for ACCSs. On the basis of their experiences, the ACCSs offered recommendations for an improved healthcare system in this regard (see Table 3).

## 4. Discussion

To our knowledge, this study is among the first to describe female ACCSs’ experiences of navigating ART and family-building options. The participants described shock at learning of their unexpected infertility risk and already compromised fertility. They grappled with intense feelings of loss and grief, along with struggles related to self-identity associated with infertility, unsuccessful ARTs, and pregnancy loss. Navigating the healthcare system related to ART and pregnancy proved difficult for all participants, as did pursuing adoption and international family-building, all compounded by the financial strain involved.

As in the existing research, ACCSs in our study reported being unaware of their risk of infertility before they were ready to begin family-building [35]. While guidelines recommend informing patients and families of the risk of infertility before cancer treatments begin, such discussions at this time can be overwhelming and difficult for patients and parents to absorb or retain [35]. Following treatment completion, infertility concerns may easily fall by the wayside because survivors might not have access to cancer-specific follow-up or care providers knowledgeable of infertility risks. Moreover, survivors themselves have reported waiting until a time in life when they were ready to have children to start investigating their fertility and parenthood options [35].

Our study highlights the need for HCPs to revisit infertility risk discussions following cancer treatment completion and regularly throughout survivorship. Given the evolving field of fertility preservation and ART, up-to-date healthcare provider knowledge appears indispensable, as well as the inclusion of fertility specialists in the care plan. Oncofertility and family-building training and communication programs for providers show promise [36,37,38]. However, these efforts might not succeed if left to individual clinicians who are already working at capacity and lack the required time and resources. Resources to directly support survivors in their decision-making about family building and preparing for potential barriers are needed, with tools such as the Roadmap to Parenthood decision aid demonstrating feasibility, acceptability, and improvements in unmet information needs, self-efficacy, and increased planning behaviors related to future family building [39]. Further, institutional guidelines, clearly articulated processes of care and referral pathways, and dedicated resources to support providers would perhaps be beneficial. Clearly, this is an area worthy of future work, particularly as it relates to our local healthcare system context.

Among our study participants, it is possible that lack of cancer follow-up, poor access to knowledgeable HCPs, and delayed consideration of family planning among survivors might have contributed to limited awareness of ART and family-building options, and the unique pregnancy and obstetrical risks associated with prior cancer treatments. While evidence specific to ART and family-building among female cancer survivors is evolving, [40] access to basic information is clearly needed by survivors and their HCPs [41]. Risks associated with pregnancy include cardiomyopathy among those treated with anthracyclines and chest RT, and miscarriage, premature birth, and low birthweight among those treated with RT to the abdomen and pelvis. [42] Although guidelines recommend that HCPs discuss these potential obstetrical risks with all female ACCSs of reproductive age before conception [42], the current structure of survivorship care in many settings, where oncological and obstetrical care remains siloed, is perhaps not amenable to support such discussions.

In our study, the mental health challenges that accompanied unexpected subfertility and infertility, pregnancy loss, and unsuccessful ART or family-building efforts were notable. In previous research, prolonged difficulties in the family-building process were experienced as a second wave of trauma after cancer, with survivors reporting shock and outrage upon learning about infertility [41]. The focus on ART and pregnancy attempts in our study was novel and highlighted worrisome levels of distress when such efforts were unsuccessful. It is well established that miscarriage can contribute to significant psychological morbidity, including anxiety, depression, post-traumatic distress disorder, and even suicide [43,44,45]. Further, several of our participants endured multiple attempts to conceive themselves or through a surrogate and later pursued alternative paths such as adoption. These cumulative losses appeared to be accompanied by marked distress spanning the many years they had spent trying to build a family of their own, with negative effects on intimate relationships and feelings of self-worth and self-identity. Further research on pregnancy loss and unsuccessful ART among cancer survivors is needed, given that much of the literature examines the impacts on those who were otherwise healthy. Our findings also suggest a dire need for psychosocial support throughout ART and family-building, though whether ACCSs have unique needs compared to the general population of individuals navigating ART remains unknown and warrants investigation.

Though a small exploratory study, our findings did highlight a diversity of unmet needs that would perhaps best be met through multidisciplinary models of care that embed oncofertility care into existing services. As of 2024, oncofertility programs were present in 56% of cancer centers and 50% of comprehensive cancer center websites in the United States. These have shown success elsewhere [26,46]. For example, a reproductive survivorship clinic model in Australia has shown promise by offering opportunities to identify and manage treatment-related reproductive risks years before family planning begins, and to provide pre-conception assessment and counseling. However, challenges with pregnancy management were also noted within this model [26].

Regardless of the model of providing oncofertility care, our findings and those of others [18] suggest that the tremendous financial burden of ART and family building faced by ACCSs deserves attention. Cancer-related financial toxicity has been found to disrupt normative young adult development, including the achievement of milestones and future goals [47], though research suggests that two-thirds of ACCSs are employed and form part of the productive population [48]. At a basic level, ACCSs ought to be proactively counseled about the potentially high costs early after completion of treatment and before starting the family-building process, given the shortened window of opportunity for ART for some and the financial implications. Financial counselling ought to equip individuals with a clear understanding of the potential costs related to fertility assessments, ART procedures, and other pathways to parenthood, such as international adoption or surrogacy, not covered by insurance or the public healthcare system. Further, advocacy to address the financial burden at a policy level is certainly warranted. Targeted efforts are needed to promote equitable access to fertility preservation and ART through financial assistance and by expanding universally accessible medical coverage to include comprehensive reproductive care for ACCSs. A recent survey of Canadian, French, and Moroccan hematologists/oncologists suggested a relationship between the offer of fertility preservation for pediatric cancer patients is proportional to public healthcare funding, such that fertility preservation is widely offered in France, where there is coverage, but not in Canada where there is no coverage (except for in the province of Quebec) [49].

## 5. Study Limitations

The study findings ought to be considered in the context of study limitations. It was difficult to recruit individuals to the study during the COVID-19 pandemic, and although we extended the study timeframe, the 15 participants likely do not represent the full spectrum of experiences of infertility treatment and family building among ACCSs in BC. Specifically, our findings do not include the experiences of ACCSs who were infertile but could not pursue fertility preservation and ART or did not pursue other family-building options. This is noteworthy considering the financial means, healthcare system knowledge, and social support that appeared to be required to pursue such options. Other research has found that breast cancer survivors with socioeconomic disadvantages were less likely to use ART [50]. Thus, our study likely underrepresents individuals in precarious socioeconomic circumstances or who are unfamiliar with navigating healthcare services. Further, our sample included women-identifying individuals who were heterosexual, with the majority Caucasian, thereby lacking diversity. Thus, our findings may not represent the experiences of individuals who are gender-diverse or not heterosexual, or from various ethnocultural groups, limiting the transferability of our findings. It is also possible that the COVID-19 pandemic influenced participants’ experiences of ART and family-building. Further, while our single-province study findings might provide insights relevant to ACCSs in other geographies, there may be greater variation in experiences considering that oncofertility care, infertility treatment, and funding coverage vary nationally and internationally.

## 6. Conclusions

Finding ways to support choice and agency in reproduction and family-building is fundamental, and a quality-of-life issue for young female cancer survivors. Although ours was a small exploratory study, participants’ accounts were powerful and illuminated profoundly challenging experiences of ART and family-building and the personal burden endured. Fertility-related matters are not always included or prioritized in the healthcare system, as our findings suggest, but are integral to health and wellbeing. Thus, there is a pressing and unmet need to more effectively integrate survivorship and oncofertility care within our healthcare context; an effort that must extend beyond the efforts of individual clinicians and be supported through dedicated time, institutional commitment, and resources. Further research is needed that articulates family-building challenges in other settings and informs the development and implementation of mitigating healthcare system approaches.

## Figures and Tables

**Table 1 curroncol-32-00369-t001:** Participant self-reported sociodemographic (*n* = 15).

Sociodemographic	*n* = 15
*Mean Age*	35.6 years
*Marital Status*	
Married/Common Law	73%
Single	20%
Divorced	7%
*Living Situation*	
Spouse/Partner	80%
Friend/Family/Roommate	13%
Alone	7%
*Place of Residence*	
Large City	73%
Small City	20%
Rural	7%
*Education Level*	
Certificate	20%
University Degree	80%
*Employment Status*	
Full time	73%
Part time	20%
Unemployed	7%

**Table 2 curroncol-32-00369-t002:** Participant self-reported medical characteristics, and ART and family-building strategies Used (*n* = 15).

	Cancer Type	Cancer Treatment	Years TTC	ART Methods Used	# of Children
1	Wilms Tumor	Chemotherapy RT Surgery	4+	IVF with Donor Oocyte/Embryo; Surrogacy	3
2	Leukemia (ALL)	Chemotherapy RT BMT	4+	IVF with Donor Oocyte/Embryo; IVF with Own Oocyte/Embryo	0
3	Lymphoma	Chemotherapy	1	Consultations	0
4	Leukemia (AML)	Chemotherapy BMT	<1	Consultations	0
5	Ewing’s Sarcoma	Chemotherapy RT Surgery	4+	IVF with Donor Oocyte/Embryo	0
6	Brain/CNS	Chemotherapy RT Surgery	4+	IVF with Donor Oocyte/Embryo; IVF with Own Oocyte/Embryo	1
7	Rhabdomyosarcoma	Chemotherapy RT Surgery	4+	IVF with Own Oocyte/Embryo; Surrogacy	0
8	Wilms Tumor	Chemotherapy RT Surgery	4+	IVF with Own Oocyte/Embryo	2
9	Lymphoma	Chemotherapy Radiation	2	IUI with sperm donor	0
10	Lymphoma	Chemotherapy	<1	Oocyte Cryopreservation	0
11	Rhabdomyosarcoma	Chemotherapy RT Surgery	<1	Consultations	0
12	Rhabdomyosarcoma	Chemotherapy RT Surgery	1	Oocyte retrievals x2	1
13	Embryonal Sarcoma	Chemotherapy Surgery	3	IVF with Own Oocyte/Embryo	0
14	Lymphoma	Chemotherapy RT	4+	IVF with Own Oocyte/Embryo	1
15	Ewing’s Sarcoma	Chemotherapy Surgery	<1	Oocyte Cryopreservation	0

Abbreviations: TTC, Trying To Conceive. ART, Assisted Reproductive Technology. RT, Radiation Therapy. ALL, Acute Lymphoblastic Leukemia. BMT, Bone Marrow Transplant. AML, Acute Myeloid Leukemia. IUI, Intrauterine Insemination.

**Table 3 curroncol-32-00369-t003:** ACCS’ Recommendations for an improved healthcare system.

Develop education materials to inform survivors of the risks of infertility at the time of treatment and throughout survivorship Develop education materials to inform healthcare providers of the risks of infertility among ACCSs
Discuss ART and family-building options with ACCSs shortly following cancer treatment completion and early in survivorship
Increase access to psychosocial services tailored to pregnancy loss, infertility, and family-building
Implement support groups for ACCSs that are age-specific (i.e., 15–25, 26+) so participants can better relate to each other
Increase the availability and affordability of ART and family-building services in Canada
Develop health services that are sensitive to pregnancy loss

## Data Availability

The data sets generated during and/or analyzed during the current study are not publicly available as participants did not consent to making their data publicly available but are available from the corresponding author on reasonable request.

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
