# Peer review of "Artificial Reproductive Technology Use and Family-Building Experiences of Female Adult Childhood Cancer Survivors: A Qualitative Study"

_curroncol, 2025, doi:10.3390/curroncol32070369_

Round 1

Reviewer 1 Report

Comments and Suggestions for Authors

The problem of oncofertility is very current and constitutes a significant challenge for both doctors and patients.

My comments are as follows:

The introduction is interesting, but too lengthy; I propose to focus on information about the risk of infertility, modern possibilities of prevention and ways of informing patients about it.

In the study group of 15 patients (according to authors)the following ART methods were used: “Eight pursued IVF with their own ovum with 3 becoming pregnant with a subsequent live birth; 5 pursued IVF with donor ovum with 1  successfully carrying a pregnancy to term; 2 pursued surrogacy and 2 pursued intrauterine insemination”. Does this mean that several (2) patients had more than one ART method used?

The analyzed group of respondents was small (15) - I propose to present in the form of a table the data concerning(separately for each survivor): diagnosis, age at the time of therapy and current age, type of therapy and the possibility/method of informing about the late effects of treatment and +/- occurrence of psychological problems

Some of the participants claim that they were not informed about the late  effects of treatment (infertility) - what was the reason? - serious condition at the time of diagnosis, young age?

In the discussion, I propose to take into account the implementation and financing of the oncofertility program in Canada and other countries with a similar economic structure.

Reviewer 2 Report

Comments and Suggestions for Authors

Dear authors, thank you for allowing me to review this insightful and timely qualitative study exploring the experiences of female adult childhood cancer survivors navigating assisted reproductive technology and family-building options. The topic is highly relevant, considering the increasing survivorship rates and the often unmet reproductive and psychosocial needs of this population.

The manuscript is well-conceived and clearly written, making a valuable contribution to the literature on survivorship care. The findings are rich and clinically relevant, with strong potential to inform practice and improve healthcare systems.

Below, I provide detailed comments:

ABSTRACT

The abstract provides a clear and concise summary of the study.
However, the phrase “towards using what was learned to inform clinical practice” could be rephrased to a more direct statement of application (e.g., "to inform improvements in clinical practice").

Additionally, the sentence 'Implications for Cancer Survivors' is slightly vague. It would be helpful to indicate the type of support planned or suggested (e.g., psychosocial, financial, informational, or integrated care).

INTRODUCTION

The Introduction is comprehensive and provides a strong rationale for the study. The background is well-documented with relevant literature.

Minor points:

The discussion of ART technologies could be slightly streamlined to avoid redundancy (lines 69-79).

The paragraph on the psychological consequences of infertility is highly relevant, but could benefit from integrating more recent literature on the long-term psychosocial impacts. Moreover, I would suggest inserting a sentence here or in the discussion to highlight the high prevalence of CCS that are employed and form part of the productive population, with the need to become parents being particularly relevant (ref. 10.3390/cancers14194586)

Consider explicitly stating the research question or objective at the end of the section to improve focus.

METHODS

The methodological approach is sound and well-documented. The use of Interpretive Description is appropriate for the research aim.

I would suggest considering these areas to strengthen. At first, reflexivity: The manuscript does not discuss how the researchers' positionality, particularly the interviewer's oncology background, may have influenced the data collection and interpretation. A brief reflexivity statement would enhance transparency. The second point of concern is the adequacy of the sample. The authors cite "information power" but do not elaborate on how this was evaluated in practice. Adding more detail on how the research team assessed that sufficient information power had been achieved would improve methodological rigour. Finally, the long recruitment period (2021-2024) raises the question of whether the COVID-19 pandemic influenced participants' experiences or the study process; this should be acknowledged.

RESULTS

The Results section is rich and well-organised, with compelling use of participant quotations. In particular, the thematic structure is logical and facilitates understanding, and the quotations are well-chosen, providing depth.

I would suggest amending Table 1, which could benefit from improved formatting (alignment issues) to enhance readability.

Moreover, I found a slight imbalance in data presentation: the section on “Grieving Loss and Redefining Identity” is significantly longer than the others. While this reflects the emotional salience, a more balanced presentation would help. Finally, in the theme “Encountering Unsupportive Healthcare”, the findings on fragmented care and inadequate provider knowledge are crucial. It would be helpful to quantify (if possible) or give a dimension on how many participants experienced these issues.

DISCUSSION

The Discussion appropriately situates the findings within the broader literature. The emphasis on the need for systematic fertility discussions post-treatment is important and well-argued. Furthermore, the discussion of psychosocial impacts is strong and resonates with previous studies.

I would suggest improving the section on practical implications, which remains somewhat general. The authors should articulate more concrete recommendations for healthcare providers and systems (e.g., specific timing for fertility discussions, integration of survivorship and fertility care, need for tailored psychosocial interventions). Second, the role of financial counselling and policy advocacy is mentioned, but could be expanded. Finally, as mentioned above, link family planning to the context of a productive working population.

The Limitations section is generally appropriate. However, the lack of diversity in the sample (all heterosexual, with a majority Caucasian) should be explicitly discussed, as this limits the transferability. As previously reported, the potential influence of COVID-19 should be mentioned. Finally, the limitation regarding not including ACCS who were unable to pursue ART or family-building options is important and could be more clearly linked to potential underrepresentation of more disadvantaged voices.

CONCLUSION

The Conclusion appropriately emphasises that fertility and family-building are core quality-of-life issues for ACCS. I found the call for further research justified, but I would suggest that the authors consider ending with a stronger statement on the urgency of improving integrated survivorship and oncofertility care, based on their findings.

I hope my comments will help you improve this interesting manuscript.

Round 2

Reviewer 2 Report

Comments and Suggestions for Authors

Dear authors, thank you for having considered all my comments and provided a point-by-point response.

I have no further comments and congratulate with the authors for having improved the overall quality of their manuscript.